# Enterococcal Phages: Food and Health Applications

**DOI:** 10.3390/antibiotics12050842

**Published:** 2023-05-02

**Authors:** Carlos Rodríguez-Lucas, Victor Ladero

**Affiliations:** 1Microbiology Laboratory, Hospital Universitario Central de Asturias, 33011 Oviedo, Spain; 2Translational Microbiology Group, Instituto de Investigación Sanitaria del Principado de Asturias (ISPA), 33011 Oviedo, Spain; 3Department of Technology and Biotechnology of Dairy Products, Dairy Research Institute, IPLA CSIC, 33300 Villaviciosa, Spain; 4Molecular Microbiology Group, Instituto de Investigación Sanitaria del Principado de Asturias (ISPA), 33011 Oviedo, Spain

**Keywords:** *Enterococcus faecalis*, *Enterococcus faecium*, antimicrobial resistance, bacteriophage, food, health

## Abstract

*Enterococcus* is a diverse genus of Gram-positive bacteria belonging to the lactic acid bacteria (LAB) group. It is found in many environments, including the human gut and fermented foods. This microbial genus is at a crossroad between its beneficial effects and the concerns regarding its safety. It plays an important role in the production of fermented foods, and some strains have even been proposed as probiotics. However, they have been identified as responsible for the accumulation of toxic compounds—biogenic amines—in foods, and over the last 20 years, they have emerged as important hospital-acquired pathogens through the acquisition of antimicrobial resistance (AMR). In food, there is a need for targeted measures to prevent their growth without disturbing other LAB members that participate in the fermentation process. Furthermore, the increase in AMR has resulted in the need for the development of new therapeutic options to treat AMR enterococcal infections. Bacteriophages have re-emerged in recent years as a precision tool for the control of bacterial populations, including the treatment of AMR microorganism infections, being a promising weapon as new antimicrobials. In this review, we focus on the problems caused by *Enterococcus faecium* and *Enterococcus faecalis* in food and health and on the recent advances in the discovery and applications of enterococcus-infecting bacteriophages against these bacteria, with special attention paid to applications against AMR enterococci.

## 1. Introduction

The discovery of antibiotics in the mid-20th century is one of the scientific advances that has had the most significant impacts on increasing life expectancy. The prescription of antibiotics and other antimicrobials to treat bacterial infections, which can often cause permanent damage and even end a patient’s life, has become a routine treatment. However, in recent decades, the misuse of antimicrobials has led to a rapid increase in the isolation of antimicrobial-resistant (AMR) bacteria. Today, AMR bacteria pose a great health problem; in Europe alone, up to 133,000 deaths in 2019 were attributable to infections caused by AMR bacteria [1], with an estimated cost for health services of over EUR 1000 million every year [2]. Seven pathogenic species were responsible for most of the deaths registered, namely, *Enterococcus faecium*, *Staphylococcus aureus*, *Klebsiella pneumoniae*, *Acinetobacter baumannii*, *Pseudomonas aeruginosa* and *Enterobacter* spp., listed together as the ESKAPE group by the World Health Organization (WHO, Geneva, Switzerland) [1,3]. Because of the great impact of these pathogens in terms of nosocomial infections, deaths and the economic losses of health services, the WHO has encouraged the scientific community to search for new ways to combat them [3]. In the context of this problem, bacteriophages have re-emerged as a potential weapon to fight AMR bacteria.

In this review, we focus on the advances in the characterization and application of enterococcal phages as rediscovered weapons against AMR *E. faecium* and *Enterococcus faecalis*.

## 2. *E. faecium* and *E. faecalis*

Enterococci are a diverse group of Gram-positive bacteria belonging to the lactic acid bacteria (LAB) group. The members of this genus are Gram-positive coccus-shaped bacteria that possess a versatile metabolism allowing them to adapt to very diverse environments and to resist rough conditions [4,5]. *E. faecium* and *E. faecalis* are the most studied species from the genus *Enterococcus* due to their role in human health [6]. These species are considered commensal bacteria of the gastrointestinal tract in mammals, including that of human beings [6,7,8]. They have also been isolated from a wide variety of environments that are mostly, but not exclusively, related to animal and human facilities, for example, cattle facilities [7], farms [9], hospitals [10] and wastewater facilities [11]. In fact, due to their persistent presence in the intestinal habitat, their robustness and endurance and the ease of their cultivation in the laboratory, enterococci are used as indicators of fecal contamination [12]. In addition, due to their presence in the gut, feces and milk of animals [13], they are also commonly found in foods of animal origin, such as meat and dairy products [11,14].

Although *E. faecium* and *E. faecalis* are considered harmless commensal bacteria, some strains are used as safe and effective probiotics, and they are present in certain cheeses in which they participate in the elimination of foodborne pathogens via the production of bacteriocins [15,16]. They can behave as opportunistic pathogens, and in recent years, they have been established as one of the major nosocomial pathogens. *E. faecalis* is considered the most pathogenic, but *E. faecium* has gained more concern due to the increasing acquisition of AMR [6,16].

### 2.1. E. faecium and E. faecalis in Food

As previously mentioned, their presence in the gut, feces and milk of mammals results in their presence in raw materials of animal origin, such as meat and dairy products. Their resistance to adverse environmental conditions allows them to grow in a wide range of pH values, temperatures and salt concentrations and to colonize foods, including fermented foods [17]. Enterococci have an ambiguous status regarding food safety. In fact, although these microorganisms belong to the LAB group (safe bacteria involved in the production of fermented foods), they have been granted with neither the Generally Regarded as Safe (GRAS) status nor the Qualified Presumption of Safety (QPS) status. Enterococci can be considered a valuable asset in cheese making, as some strains can be used as adjunct starter cultures [18,19]. The role of enterococci during cheese making is based on the large variety of technologically interesting enzymatic activities, such as protease, peptidase and lipolytic activities, which contribute to the organoleptic properties during the maturation process [18]. In addition, some strains are able to produce enterocins—so-called bacteriocins—and these produced by different strains of enterococcus that can inhibit the growth of several foodborne pathogens, such as *Staphylococcus aureus*, *Listeria monocytogenes* and *Salmonella enterica* [20]. However, their presence in foods has also been associated with the production of biogenic amines (BAs), toxic compounds that can cause food poisoning [21,22]. In fact, *E. faecalis* has been identified as the main species responsible for the accumulation of elevated concentrations of tyramine and putrescine [23,24,25], two of the most frequent BAs in dairy products [26,27]. Moreover, due to the increase in recent years of multidrug-resistant (MDR) bacteria, including enterococci, and the coining of the One Health concept to prevent their proliferation [28], concern regarding MDR enterococci in food has emerged [29]. These AMR enterococci can reach the food chain, where they can be transmitted directly or indirectly to humans and act as reservoirs of AMR genes that can be transferred to human-adapted strains or to other pathogenic bacteria [29]. Of special concern is the increase in vancomycin-resistant enterococci (VRE) from different food sources [30,31,32,33]. Thus, although the presence of enterococci in food could be considered beneficial in some scenarios, in general, they are considered a potential health threat.

### 2.2. E. faecium and E. faecalis in Human Health

Enterococci are considered a commensal organism of the human gastrointestinal (GI) tract, and they can be found in the GI microbiota of more than 90% of healthy people [34]. The first description of an enterococcal human infection was in 1899, when MacCallum and Hastings reported infective endocarditis (IE) caused by a bacterium that they called *Micrococcus zymogenes*, later identified as a member of the *Enterococcus* genus [35,36]. Enterococci were subsequently shown to be the cause of several kinds of infections, both community- (e.g., including pelvic infections, urinary tract infections and IE) and healthcare-associated infections (HAIs) (e.g., including surgical site infections, and urinary and bloodstream catheter-related infections) [37]. Therefore, enterococci usually display low levels of virulence, but they can also act as an opportunistic pathogen, causing severe infections, mainly in vulnerable patients, such as those who are immunocompromised, have undergone invasive procedures (e.g., the insertion of urinary or blood catheters) and have previously received antimicrobial treatments [34].

#### 2.2.1. Epidemiology of Enterococcal Infections

*E. faecium* and *E. faecalis* account for more than 90% of the enterococci recovered from clinical samples in humans. Among them, *E. faecalis* is the most frequent species (80–90%) causing human infection, followed by *E. faecium* (5–10%) and other species (less than 10%) [38]. In recent decades, enterococci have become a first-rate clinical problem, being one of the most common microorganisms of HAIs around the world [39,40]. Several factors related to the host and to the microorganism have contributed to this conversion of a commensal pathogen into one of the major causes of HAIs. The most relevant factors associated with enterococci are their intrinsic resistance to some antimicrobials (e.g., aminoglycosides, cephalosporins and clindamycin); their ability to acquire and disseminate AMR determinants (e.g., linezolid and vancomycin resistance); and the plasticity of their genome, which may contribute to improve their adaptation to harsh environments. Moreover, the increasing number of patients undergoing immunomodulatory therapies, undergoing invasive procedures or receiving multiple antimicrobial treatments, all of which are factors associated with the host, favors the role of enterococci to cause disease [34,41].

#### 2.2.2. Antimicrobial Resistance in Enterococci

##### Resistance to ß-Lactams

Enterococci are intrinsically resistant to cephalosporins, and they present a natural reduced susceptibility to penicillin due to the expression of low-affinity penicillin binding proteins (PBPs), designated PBP4 in *E. faecalis* and PBP5 in *E. faecium* [34,42]. Moreover, many enterococci strains show tolerance to the bactericidal activity of ß-lactams, with the minimal bactericidal concentrations being higher than the minimum inhibitory concentrations (MICs) [5]. This situation can be solved with the addition of an aminoglycoside (typically streptomycin or gentamicin) to an active ß-lactam, which results in bactericidal synergism [43,44].

A higher-level resistance to penicillin or ampicillin resistance in enterococci can be due to the overexpression of chromosomal PBP4 and PBP5 in *E. faecalis* and *E. faecium*, respectively, or through acquired mechanisms [45,46,47]. The former is anecdotic, as acquired mechanisms are the most frequent cause of ampicillin resistance. Acquired mechanisms include ß-lactamase production and mutation acquisition in low-affinity PBP4 and PBP5 [48,49]. Currently, ampicillin resistance in enterococci is mainly mediated by the acquisition of mutations in PBP, and it is far more prevalent in *E. faecium* than in *E. faecalis* [49]. Ampicillin-resistant *E. faecium* due to acquired mutations in the PBP5-encoding gene has been linked to a hospital-associated (HA) clade, and it emerged in the late 1970s in the United States (US) [34]. Today, there is a high rate of ampicillin resistance in *E. faecium* strains, and it exceeds 70% in many countries [5,50].

##### Resistance to Aminoglycosides

Enterococci are intrinsically resistant to clinically achievable concentrations of aminoglycosides due to the poor penetration of these agents through the bacterial cell wall in *E. faecalis* and due to two chromosomally encoded genes, namely 6′-*N*-aminoglycoside acetyltransferase (*aac(6′)-Ii*) and rRNA methyltransferase (*efmM*) in *E. faecium* [5,34]. As previously mentioned, this type of resistance can be overcome with the addition of an agent that disrupts cell wall synthesis, such as ß-lactams. Some strains can also exhibit a high level of aminoglycoside resistance (MIC > 500 mg/L for gentamycin and MIC > 2000 mg/L for streptomycin) through the acquisition of aminoglycoside-modifying enzymes (phosphotransferases, acetyltransferases and nucleotidyltransferases), which inhibit the aforementioned synergic effect [5,51].

##### Resistance to Glycopeptides

Vancomycin, the main member of the glycopeptide family, was the first-line treatment of ampicillin-resistant *E. faecium* for decades, without reports of VRE strains until the 1980s [52,53,54]. Glycopeptide resistance in enterococci is mediated by the acquisition of eight different genes of the *van* operon (*vanA*, *vanB*, *vanD*, *vanE*, *vanG*, *vanL*, *vanM* and *vanN*). Moreover, *E. casseliflavus* and *E. gallinarum* exhibit intrinsic low-level resistance to glycopeptides through the presence of a *vanC* gene in their chromosome [55,56]. These genes code for the terminal amino acids of peptidoglycan precursors different from the original form (D-Ala-D-Ala). Thus, the modified amino acids D-Ala-D-Lactate and D-Ala-D-Serine present a lower affinity to glycopeptides, leading to high-level and low-level resistance to glycopeptides, respectively [57]. The *vanA* and *vanB* genes are the main mechanism of resistance to glycopeptides in enterococci, mainly being present in *E. faecium* [55,58]. The prevalence of glycopeptide-resistant *E. faecium* varies widely between continents and countries. Accordingly, the percentage of resistance to glycopeptides in *E. faecium* invasive isolates is more than 60% in the US, 37% in Australia and 16.8% in European countries (with national percentages ranging from 0.0 to 56.6%) [49,50,59,60,61]. HA ampicillin-resistant *E. faecium* clones often acquire resistance to glycopeptides, highlighting the importance of *E. faecium* as a nosocomial pathogen [62,63]. The scarce active antimicrobials available to treat infections caused by this MDR microorganism are a global cause for concern.

##### Resistance to Linezolid

Although linezolid resistance in enterococci remains uncommon, the number of linezolid-resistant enterococci (LRE) has increased in recent years. The main mechanism of linezolid resistance in Gram-positive bacteria is point mutations in the central loop of domain V of the *23S rRNA* gene, among which the G2576T (*Escherichia coli* numbering) nucleotide mutation is the most described [64,65]. Other point mutations in the genes *rplC*, *rplD* and *rplV*, which code for the L3, L4 and L22 ribosomal proteins, respectively, are also associated with a decreased susceptibility to linezolid; however, they play a minor role [66,67]. Moreover, the acquisition and dissemination of transferable linezolid resistance genes, namely, *cfr*-like, *optrA* and *poxtA* genes, have been increasingly reported in linezolid-resistant Gram-positive bacteria in recent years [10,67,68,69,70]. The *cfr*-like genes encode a 23S rRNA methyltransferase, which confers resistance to phenicols, lincosamides, oxazolidinones, pleuromutilins and streptogramin A (PhLOPS_A_ phenotype) [71,72]. However, the *optrA* and *poxtA* genes code for the ribosomal protection proteins of the ABC-F family, and they confer resistance to oxazolidinones and phenicols, as well as tetracyclines in the case of *poxtA* [73,74]. Nowadays, the *cfr*, *cfr(B)*, *cfr(D)*, *optrA* and *poxtA* genes have been described among enterococci from different sources (animal, human and environmental samples) and countries [75]. The main linezolid resistance mechanisms are mutations in the 23S rRNA in *E. faecium* and the *optrA* gene in *E. faecalis* [68]. The spread of these transferable linezolid resistance genes to difficult-to-treat bacteria, such as VRE, is a cause for concern. Unfortunately, outbreaks caused by *E. faecium* strains that are resistant to vancomycin and linezolid (*optrA*-positive) have already been reported [76,77].

##### Resistance to Daptomycin

Daptomycin-resistant enterococci (previously called daptomycin-nonsusceptible enterococci) are uncommon, and they have often been associated with prior exposure to the drug [67]. Daptomycin resistance in enterococci is mainly mediated by structural alterations of the cell envelope through a variety of mutations, mainly in the three-component regulatory system LiaFSR [5,78]. This alteration of the cell envelope produces a repulsion of daptomycin from the membrane. Moreover, daptomycin resistance in *E. faecium* is also associated with mutations in the *cls* gene [49]. Daptomycin resistance is more common in *E. faecium* than in *E. faecalis*, which is probably related to the use of this drug to treat vancomycin-resistant *E. faecium* infections [78].

## 3. Bacteriophages of *E. faecium* and *E. faecalis*

Bacteriophages, or phages, have emerged in recent years as a potential bioweapon to combat MDR bacteria [79,80]. Phages are viruses that infect and kill bacteria. They are the most abundant entities on Earth and the most genetically diverse biological entities due to their mosaic genome structure and ability to mutate and recombinate [81,82]. In addition, they are ubiquitous in all types of environments, from the sea to the human gut [83]. As they are natural predators of bacteria, they have been suggested to be one of the most promising alternative therapeutic agents against MDR bacterial infections [79]. The use of phages as therapeutic agents (phage therapy) was suggested immediately after their discovery in the early XX century by Frederick Twort and Félix d’Herelle. However, the discovery of antibiotics, with a broader spectrum of action, meant that their use declined rapidly [79], except for in some countries in Eastern Europe and in the former Soviet Union where phage therapy was active, as in the Eliava Institute in Georgia [84], a reference center for phage therapy worldwide. The global problem of MDR bacteria and their consequences in terms of lives and health system costs [2] have led to a renewed interest in the study of phages and their application in phage therapy. Moreover, under the umbrella of phage therapy, the use of phages has been proposed in other fields, including food safety [85,86,87].

In this context, in recent years several enterococcal-infecting phages have been isolated and characterized—genetically and functionally (Appendix A). It is remarkable that the number of *E. faecalis*-infecting phages that have been characterized is higher than that of *E. faecium*-infecting phages [88], as, in the last year, the number of *E. faecium* phages has increased. In Appendix A, we can see that 101 genomes of *E. faecalis*-infecting phages are available, whereas only 16 of *E. faecium* can be found. Whether or not this bias is related to abundance, the ease of isolation under laboratory conditions or different searching pressures is unclear. It is astonishing that although there is a large number of molecular techniques available for the precise identification of bacterial species, there is still a large number of phage genomes (24) identified as infecting *Enterococcus* spp. (Appendix A). However, it is remarkable that some phages are able to infect strains of both species, that is, *E. faecalis* and *E. faecium* [89,90,91]. This could be considered an advantage if a general phage cocktail designed to treat enterococcal infections is intended.

The *Enterococcus*-infecting phages that have been isolated to date are taxonomically widely diverse, as there are phages belonging to eleven different genera from four families (Figure 1; Appendix A). Regarding the genus, the most abundant genera, accounting for almost half of the phages, are *Efquatrovirus* (represented by 39 phages) and *Saphexavirus* (represented by 25 phages), both belonging to the *Siphoviridae* family. Siphoviridae is the most abundant morphology, with three times more isolates than the Myoviridae and Podoviridae morphologies. The genome size distribution has a wide range, from approximately 18 kbp to 150 Kbp, but this heterogeneity is mostly related to taxonomic differences rather than genome diversity. Small genomes are typical of the *Rountreeviridae* family (previously known as *Podoviridae*), whereas large genomes are characteristic of the *Herelleviridae* family (previously known as *Myoviridae*) [92,93]. Nevertheless, within the same genera, heterogeneity in the genome size is observed, thus indicating differences among their members (Appendix A).

Most of the reported enterococcal-infecting phages are virulent, at least those included in databases as single entries. In general, temperate phages are described as prophages that are identified and characterized as part of an analysis of one strain or isolate genome [94]. This is linked to the fact that temperate phages are not a good option for phage therapy due to their known involvement in the phenomenon of horizontal gene transfer, one of the main mechanisms involved in the spread of virulence and AMR genes [95]. In addition, after entering a new infecting cell, if a temperate phage undergoes the lysogenic cycle, the infected bacteria do not die; they can continue to spread and become resistant to infection by that same phage. Nevertheless, in some cases, if no alternative exists, temperate phages can be converted into virulent ones by selecting or constructing mutant phage variants [96,97], as has been achieved in the case of the *E. faecalis* ΦEf11 prophage, resulting in an additional increase in host range and progeny [98]. As previously mentioned, the lifestyle (virulent or temperate) of the described phages could be biased by the fact that this is an exclusion criterion for phage therapy due to their putative role in the transference of antimicrobial resistance genes [95]. In this sense, the absence of these genes is also a requirement. Most probably, this is related to the lack of such genes in the genomes of the described phages.

Although enterococci have been documented in many different ecosystems, most of the reported isolation sources are sewage and wastewater (Appendix A). This could be related to the role of *Enterococcus* as an indicator of fecal contamination in water [12]. Nevertheless, other sources, in the search for increased phage diversity or specific target applications, have been assayed, such as human stools [89] and cheese [87,88] (Appendix A).

## 4. Food Applications of Enterococcal Bacteriophages

As mentioned in a previous section, *Enterococcus* plays a yin–yang role in foods, where it can be considered a beneficial player responsible for the accumulation of toxic compounds or reservoirs of AMR genes.

*E. faecalis* has been identified as the main cause of the accumulation of tyramine [23,99] and putrescine [24]—together with *Lactococcus lactis* [100]—in dairy products, one of the food matrixes in which BAs can reach the highest concentration [27,101]. Strategies for the reduction of BAs in food have been proposed, for example, eliminating BAs after they have formed and accumulated via the addition of BA-degrading microorganisms [102,103,104] or reducing the number of BA-producing microorganisms via different treatments, such as the use of pasteurization or high-pressure technologies [105,106], with the latter being the technical process most employed during food production. However, these methods work by generally reducing the microorganisms present in the food matrix, thus affecting other bacteria that participate in the fermentation process. As enterococci belong to the LAB group, the methodologies applied to reduce their presence also act on other LAB, affecting the development of the organoleptic characteristic of the final product. Thus, tailored measures targeting only the *Enterococcus* population are needed. In this sense, phages infecting *E. faecalis* have been proposed as highly specific tools to reduce the content of BAs in dairy products [88]. The *E. faecalis* Q69 phage has been applied to reduce the presence of tyramine, one of the most toxic BAs found in cheese [107,108], in an experimental cheese model [87]. The phage was added directly to milk [multiplicity of infection (MOI 0.1)] used for cheese making, and after 60 days of ripening, reductions in tyramine concentrations of about 85% were achieved [87], and most importantly, the concentration of tyramine was reduced below the safety threshold level proposed [109]. In another assay, the *E. faecalis*-infecting phage 156 was applied (MOI 0.1) to reduce tyramine and putrescine, another toxic BA frequently found in cheese [21,26,110], taking advantage of the fact that *E. faecalis* is responsible for the accumulation of elevated concentrations of both BAs. In this case, significative reductions in tyramine and putrescine of 95% and 77%, respectively, were achieved [111] after 60 days of ripening. Interestingly, both phages showed the ability to control the population of *E. faecalis* from the early stages of cheese making, and both were partially resistant to the pasteurization process, allowing for both technologies to be applied if desired. The fact that some of the phages proposed as a tool to reduce the content of BAs in dairy products can infect MDR enterococci, including VRE [88,111], suggests that bacteriophages could also be applied to reduce the presence of MDR enterococci in food [112], thus contributing to the One Health strategy’s aim of reducing the amount of MDR bacteria in the environment.

Biofilms in the food industry also present a food safety threat, as they can act as reservoirs of foodborne spoilage or pathogenic bacteria [113,114], including BA-producing ones [115]. Different phages have been described as potential tools to eliminate biofilms formed by *E. faecalis* or *E. faecium* [91,116,117,118]. Although the great potential of enterococcal phages as biofilm elimination agents in food facilities surfaces, to the best of our knowledge, there are no reports regarding this interesting application, thus opening the opportunity for further research.

## 5. Human Health Applications of Enterococcal Bacteriophages

The continued increase in AMR among enterococci and their abilities to form biofilms and survive in harsh environments have been related to poor clinical outcomes in some cases [119]. Different studies have reported the use of enterococcal phages in the treatment of *Enterococcus* infections through in vitro and in vivo studies, including the use of biofilm, root canal and animal models. However, the use of phage therapy in human patients is limited to some case reports, and, at present, no clinical trials are being carried out.

### 5.1. In Vitro Models

#### 5.1.1. Biofilm Models

The use of a single phage, a phage cocktail or their combination with antimicrobials in the treatment of bacterial biofilms has been explored in several studies [120,121]. Biofilm-associated infections are related to poor microbiological and clinical outcomes when antimicrobials are used. Phage therapy, in some cases, has been proven to be more effective against MDR biofilm infections than antimicrobials [91]. Several studies have reported the ability of different phages to infect and disrupt biofilms [120,121]. Anti-biofilm activity is mainly tested using microtiter plates with the crystal violet method and confocal laser scanning microscopy [121]. Using these methodologies, the anti-biofilm activity of several phages belonging to different families (*Herelleviridae*: vB_EfaH_EF1TV; *Siphoviridae*: Efa02, EfaS-SRH2, SHEF2 and vB_EfsS_V583; and *Podoviridae:* vB_ZEFP) has been demonstrated against *E. faecalis* [119,122,123,124,125]. Moreover, phage therapy has been shown to be a potential weapon against the biofilms formed by MDR *Enterococcus*. In a previous study, it was found that the EFDG1 phage was able to infect vancomycin-resistant *E. faecium* and *E. faecalis* strains. Moreover, EFDG1 significantly reduced a 2-week-old biofilm formed by a vancomycin-resistant *E. faecalis* strain V583 [91]. Furthermore, the vB_EfsS_V583 phage inhibited the biofilm formation by a vancomycin-resistant *E. faecalis* strain for 7 days. However, a poor ability to eradicate mature biofilms was revealed, with significant disruption only being observed in 1- to 2-day-old biofilms [124].

Some authors have developed models closely resembling biofilm formation during in vivo infections to avoid the possible limitations of microtiter plate studies. Thus, El-Atrees et al. studied the effects of the EPA, EPC and EPE phages on *E. faecalis* (EF104, EF134 and EF151) adherence to urinary catheter surfaces [126]. Their findings proved the ability to prevent biofilm formation by reducing the number of cells adhering to the catheter surface to a range of 30.8–43.8%. Moreover, they were also able to eradicate the number of cells pre-adhering to the catheter surface to a range of 48.2–71.1%. Nevertheless, when the anti-biofilm activities of the same phages were evaluated on microtiter plates, they showed more efficacy in both the prevention of biofilm formation and the eradication of the preformed biofilm, achieving ranges of 38–39.9% and 71–78.4%, respectively [126]. In a similar study, silicone Foley catheters were covered with an *E. faecalis* biofilm, and then they were exposed to the vB_EfaS-271 phage for 3, 6 or 24 h. A significant decrease in the number of viable *E. faecalis* cells was observed after three hours when a higher MOI was used. However, lower MOI ratios needed a longer time (6 h) for considerable effects to be observed. Unexpected results were observed at 24 h, with a large number of *E. faecalis* cells surviving in samples treated with 10 MOI compared to those treated with 0.0001 or 0.01 MOI. The authors speculated that a greater selection of phage-resistant mutants could occur under high-MOI conditions. However, such mutants seemed to be less competitive than wild-type cells [127]. In another approximation of biofilm formation during in vivo infections, Melo et al. developed an in vitro collagen wound model (CWM) of biofilm formation with two phages: vB_EfaS-Zip (*Rountreeviridae*) and vB_EfaP-Max (*Siphoviridae*). Both phages showed lytic activity against *E. faecium* and *E. faecalis.* In the CWM, vB_EfaP-Max and vB_EfaS-Zip were able to reduce the number of viable cells of *E. faecalis* and *E. faecium*, respectively, during the first eight hours. However, in both cases, the number of cells in the control and phage-treated biofilms in the CWM was similar at 24 h. In a new CWM, a cocktail comprising the two phages was used to infect a dual-species (*E. faecium* and *E. faecalis*) biofilm. In this last assay, a statistically significant reduction in the concentrations of the cells in the treated biofilms was observed at 3, 6 and 8 h compared to those in the control, and a residual reduction was also detected at 24 h. The emergence of phage resistance might be related to the loss of or a reduction in anti-biofilm activity when a phage alone or a cocktail phage is applied, respectively [90]. Although the selection of phage-resistant mutants is an issue in phage therapy, in some cases, this selection may be an opportunity. Liu et al. described the strong lytic activity of the EFap02 phage against the *E. faecalis* strain Efa02 and identified the glycosyltransferase gene Group 2 (*gtr2*) as its receptor. Unfortunately, the rapid emergence of phage-resistant mutants was observed by the authors. The phage-resistant strain EFa02R had loss-of-function mutations in the *gtr2* gene, responsible for the biosynthesis of capsular polysaccharides. Not only does the loss of receptors in EFa02R prevent phage adsorption, but it also impairs the biofilm formation ability of these mutants. Therefore, capsular polysaccharide loss could revert the inactivation of some antimicrobials caused by the biofilm [125].

A phage alone or a phage cocktail combined with antimicrobials could prevent the emergence of phage resistance and enhance their activities separately [128]. Phage–antimicrobial synergy has been reported, even against bacteria resistant to the antimicrobial used in the combination [129]. Similar to this, daptomycin plus a phage cocktail (113 and 9184) showed synergic bactericidal activity against daptomycin-nonsusceptible *E. faecium* [130]. Likewise, phage–antimicrobial synergy against vancomycin-resistant *E. faecalis* V583 was also observed when a combined treatment of vancomycin and the EFLK1 phage was applied. This combination was able to reduce viable bacterial counts by nearly 8 logs in a well-established biofilm, whereas treatment with the phage alone only achieved a reduction of 4 logs, and vancomycin alone failed entirely [131].

#### 5.1.2. Human Root Canal Model (Ex Vivo)

*E. faecalis* is frequently detected in asymptomatic and persistent endodontic infections, with prevalence ranging from 24 to 77% [132]. Its ability to invade the dentinal tubes of the root canal walls of human teeth, their aforementioned ability to form biofilms and their ability to survive in harsh environments allow this microorganism to cause persistent infections, and it is difficult to treat them [121,132,133]. The treatment of these infections includes mechanical debridement and chemical agents, such as chlorhexidine and sodium hypochlorite (NaOCl), which are generally effective [132,133]. However, several treatment failures have been reported, and therefore, the development of new therapeutic alternatives is necessary [132]. In this context, enterococcal phage therapy has been studied in ex vivo models of root canal infection. In a previous study, it was found that the EFDG1 phage was able to prevent *E. faecalis* root canal infection in an ex vivo model performed with human-extracted teeth [91]. Likewise, the vB_ZEFP phage showed a greater ability to reduce bacterial leakage from the root apex than other treatments (NaOCl and NaOCl plus EDTA) [123]. Similar studies have demonstrated the efficacy of different phages to destroy *E. faecalis* biofilms in root canal systems: vB_Efa29212_2e (*Siphoviridae*), vB_Efa29212_3e (*Herelleviriae*) and vB_EfaS_HEf13 (*Siphoviridae*) [134,135,136,137]. With regard to MDR *Enterococcus*, Tinoco et al. evaluated the activity of ΦEf11/ΦFL1C(∆36)P^nisA^, an engineered phage, against vancomycin-resistant *E. faecalis* V583 in an ex vivo model of root canal [138]. The treatment with the phage generated a reduction of 99% for the V583-infected models. In contrast, a scarce reduction of 18% was observed in biofilms formed by the *E. faecalis* JH2-2 strain (a fusidic acid- and rifampicin-resistant, vancomycin-susceptible strain) [138].

### 5.2. In Vivo Models

In vivo studies using animal models are essential to evaluate the safety and efficiency of new therapies, including phage therapy. Among them, the most frequently used include models performed in *Galleria mellonella*, zebrafish embryos and mice [139]. The *G. mellonella* animal model has previously been used to assess the virulence of VRE [89]. Although this model is cost-effective in the evaluation of the potential of phage therapy, to date, only two studies have assessed the efficacy of phage therapy against larvae infected by *Enterococcus* [89,140]. In the first study, the administration of a phage cocktail comprising the MDA1 (*Rountreeviridae)* and MDA2 (*Herelleviridae*) phages was effective against larvae infected with a vancomycin-resistant *E. faecium* strain (VRE004). The larvae were injected with 10^7^ colony-forming units (CFU)/10 μL of the VRE004 strain, and two groups were employed. These groups were injected with the phage cocktail at a concentration of 2 × 10^6^ plaque-forming units (PFU)/10^6^ μL, one of them 1 h prior to (the prophylactic group) and the other 1 h after (the treatment group) the VRE injection. After 48 h of follow-up, both groups demonstrated efficacy, being 3.7 (the treatment group) and 6.5 (the prophylactic group) times more likely to survive than the larvae injected with VRE only [140]. In the second study, the activity of the phage vB_EfaH_163 (*Herelleviridae*) against larvae infected with a vancomycin-resistant E. *faecium* (VRE-13) strain was studied. The larvae were injected with the VRE-13 strain at a concentration of 10^5^ CFU/larva. After 1 h, the larvae were injected with PBS (the control group) or a phage suspension at an MOI of 0.1, and the number of deaths was monitored for five days. Treatment with the vB_EfaH_163 phage increased larval survival by 20% compared with the control group, although no statistically significant differences were observed [89]. In another assay, the therapeutic potential of phage SHEF2 (*Siphoviridae*) in treating systemic *E. faecalis* infections in an in vivo zebrafish embryo infection model was studied. In this model, the zebrafish embryos were infected with *E. faecalis* (OS16 strain), and two hours later, they were injected with SHEF2 or a heat-killed sample of SHEF2 at an MOI of 20 (with respect to the *E. faecalis* inoculum). The zebrafish infected with OS16 alone or with heat-killed SHEF2 showed a mortality rate of 73%, whereas those injected with SHEF2 showed a mortality rate of 16% (*p* < 0.0001) [119]. Nowadays, a number of studies are evaluating enterococcal phage therapy against different kinds of *E. faecalis* infection using mice models. Enterococcal phage therapy has been shown to be a promising candidate for the treatment of *E. faecalis* endophthalmitis (a rare cause of postoperative infection) in mice models. Thus, Kishimoto et al. demonstrated a decrease in the number of viable bacteria and the infiltration of neutrophils in mice eyes infected with vancomycin-susceptible and -resistant *E. faecalis* when they were treated with phages [ΦEF24C-P2, ΦEF7H, ΦEF14H1 and ΦEF19G (*Herelleviridae*)] [141,142]. Several studies have evaluated the efficacy of phage therapy in an *E. faecalis* sepsis mice model using intraperitoneal injections. The intraperitoneal administration of phage IME-EF1 or its endolysin, at a 10 MOI, 30 min after *E. faecalis* 002 inoculation resulted in survival rates of 60% and 80%, respectively [143]. A single injection of 3 × 10^8^ PFU of the phage ENB6, administered 45 min after a vancomycin-resistant *E. faecium* (CRMEN 44) challenge, was able to rescue 100% of the mice [144]. A single intraperitoneal administration of other phages (at different doses and times following bacterium inoculation) was also enough to protect all the mice infected with enterococcus (including VRE): ΦEF24C (MOI 0.01/20 min), EF-P29 (4 × 10^5^ PFU/1 h), SSsP-1 and GVEsP-1 (3 × 10^9^ phage stock/3 h) and a phage cocktail (comprising the phages EFDG1 and EFLK1) (2 × 10^8^ PFU/0 h and 1 h) [145,146,147,148].

### 5.3. Phage Therapy in Humans

To date, only a few case reports have described the use of phage therapy against enterococcus infections in humans (Table 1) [149,150,151,152,153]. Three patients suffering from chronic bacterial prostatitis caused by *E. faecalis*, previously unsuccessfully treated with long-term targeted antimicrobials, autovaccines and laser bio-stimulation, were selected for phage therapy. Phage treatment was rectally applied, twice daily, with 10 mL of bacterial phage lysate (with a phage titter between 10^7^ and 10^9^ PFU/mL) for 30 days. Encouraging results were obtained in the three patients regarding bacterial eradication, the abatement of clinical symptoms and the lack of early disease recurrence [152]. In another case of chronic bacterial prostatitis, this time polymicrobial (with different staphylococcal species, *E. faecalis* and *Streptococcus mitis*), three phage preparations from Eliava Institute (Pyo, Intesti and Staphylococcal Bacteriophage preparations), with an approximate phage titter between 10^5^ and 10^7^ PFU/mL, were used. These preparations were applied via three routes: the oral route (20 mL of Pyo and Intesti Bacteriophage per day for 14 days), the rectal route (Staphylococcal Bacteriophage suppositories twice a day for 10 days) and the urethral route (Intesti Bacteriophage instillations once a day for 10 days). A significant improvement in symptoms was observed after phage therapy, and the patient was considered in full remission [149]. A commercial preparation of Pyo Bacteriophage (with an unknown phage titter) (Eliava Institute, Tbilisi, Georgia) was also applied in a case of a recurrent femur osteomyelitis infection after multiple failed medical and surgical therapy regimens. The infected bone was rinsed with 40 mL of the phage solution after the debridement surgery (intraoperative), followed three times per day with 10–20 mL for seven days using a draining system (the draining system was closed to allow a contact time of 10 min). The patient was concomitantly treated with amoxicillin for three months. A follow-up of the patient after eight months showed no signs of clinical or radiological recurrence, and the patient was considered infection-free [151]. Pyo and Intesti Bacteriophages (with an unknown phage titter) were also used in a case of an *E. faecalis* hip prosthetic joint infection. In this case, the phages were administered orally (10 mL), with the first one administered in the morning and the second one administered in the evening, for two periods of 19 days with a pause of 2 weeks between both periods. The patient was treated concomitantly with amoxicillin in the first phage treatment period and doxycycline in the second one. Antimicrobial therapy was suspended after a final course of doxycycline for four months. Over the next two years, the patient recovered and had no hip complaints [153]. Lastly, Paul et al. reported the first case of a vancomycin-resistant *E. faecium* abdominal infection treated with intravenous injections of cocktail phages (comprising the EFgrKN and EFgrNG phages, with a joint titter between 10^7^ and 10^8^ PFU/mL). Phage therapy was intravenously administered (2 mL/Kg/12 h over 2 h) for 20 days in a one-year-old girl, critically ill, needing three successive liver transplants with a persistent vancomycin-resistant *E. faecium* infection. Although the disease course was complex, the authors linked the clinical improvement to the phage application [150].

As can be seen in these cases, the use of phage therapy carried out on a compassionate-use basis has provided encouraging results in the healing of patients who suffered from difficult-to-treat infections and did not have other alternative treatment options. To date, clinical trials have assessed the effect of phage therapy on a few pathogens or different types of infections (e.g., urinary tract infections) with discordant results, but no clinical trials focusing on Enterococcus have been performed [120,154]. The safety and feasibility of phage therapy, including when it is administrated intravenously, have been demonstrated in several reports [155,156]. Therefore, the development of double-blind randomized clinical trials is necessary to assess the true efficacy of phage therapy in human health. Moreover, these studies must answer some questions and assess the contribution of various factors to the outcome of phage therapy, such as the quality of phage preparation; their titter, dosage and route of administration; and the concomitant use of antimicrobials.

## 6. Conclusions

AMR in pathogenic microorganisms is a major global threat and a leading and increasing cause of mortality worldwide. The interest in the possibilities of phage therapy as a new weapon to fight AMR microorganisms has boosted research on bacteriophages, particularly that on *Enterococcus*-infecting phages. As shown in this review, there are a large number of characterized phages that fulfill the requirements for application in phage therapy and feed. However, the lack of legislation regulating their use in food limits their possible application in most countries. Similarly, the lack of procedures, as well as the lack of a definition of good production practices for phage suspensions, limits their application to compassionate use in patients for whom there is no other treatment alternative. Thus, a step forward is still needed to standardize procedures that allow for their systematic use in practical clinical applications beyond compassionate use.

## Figures and Tables

**Figure 1 antibiotics-12-00842-f001:**
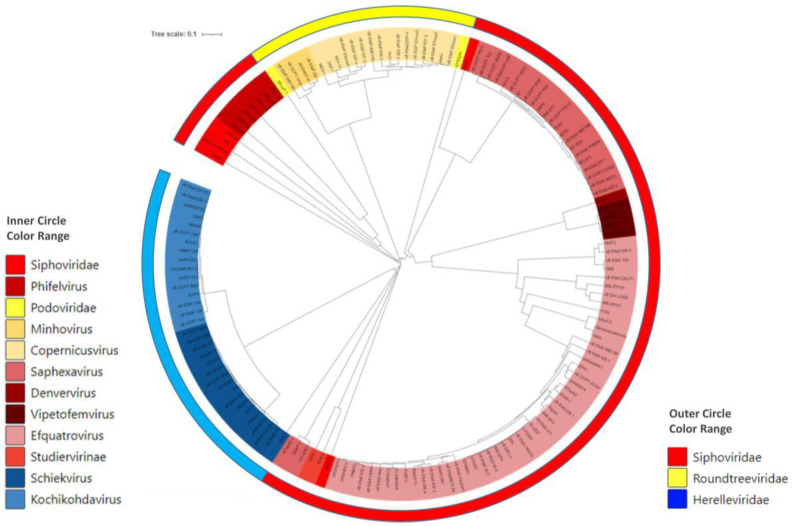
Phylogenetic tree of *Enterococcus*-infecting bacteriophages based on the major capsid proAppendix A. The tree was generated by using the unweighted pair group method with arithmetic means (UPGMA) and by employing MAFFT v.7 software (https://mafft.cbrc.jp/alignment/server/ (accessed on 12 January 2023)). The generated phylogenetic tree was visualized using the iTOL web server (https://itol.embl.de/ (accessed on 7 March 2023)).

**Table 1 antibiotics-12-00842-t001:** Characteristics of available studies of phage therapy against *Enterococcus* in humans.

Type of Infection and No of Subjects (n)	Target Strain	Phage	Application Route	Concomitant Antimicrobial Use	Outcomes	Reference
Chronic bacterial prostatitis (n = 3)	*E. faecalis*	No data	Rectal	No	Bacterial eradication Abatement of symptoms Lack of early disease recurrence	[146]
Chronic bacterial prostatitis (n = 1)	*E. faecalis* ^a^	Pyo ^b^, Intesti ^b^ and Staphylococcal bacteriophage ^b^	Oral, rectal and urethral	No	Bacterial eradication Significant improvement in symptoms	[147]
Femur osteomyelitis (n = 1)	*E. faecalis*	Pyo bacteriophage ^b^	Direct rise of the infection site	Yes (amoxicillin)	No signs of clinical or radiological recurrence	[148]
Hip prosthetic joint infection (n = 1)	*E. faecalis*	Pyo ^b^ and Intesti bacteriophage ^b^	Oral	Yes (amoxicillin and doxycycline)	Not hip complaints	[149]
Intrabdominal infection (n=1)	VR *E. faecium*	EFgrKN and EFgrNG	Intravenous	Yes (linezolid)	Clinical improvement	[150]

VR: vancomycin-resistant. ^a.^ Polymicrobial infection which include members of the Staphylococcal species (including *S. aureus*), *Streptococcus mitis* and *E. faecalis*. ^b.^ Standard phage preparations made by Eliava Institute (Tbilisi, Georgia).

## Data Availability

Data are contained within the article.

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
