# Peer review of "Enterococcal Phages: Food and Health Applications"

_antibiotics, 2023, doi:10.3390/antibiotics12050842_

Round 1

Reviewer 1 Report

Please find my comments in the attached file.

Author Response

Remarks to the authors

This review article provides an overview of the potential use of bacteriophages to combat multidrug-resistant bacteria in food and health, with a focus on enterococcal infecting phages. The authors discussed the mechanisms for resistance to various traditional antimicrobial agents in Enterococcus bacteria. The author reviewed different in vitro, ex vivo and, in vivo studies that have explored the effectiveness of phages in treating Enterococcus biofilms and infections. Finally, the authors briefly described several case reports of phage therapy used to treat bacterial infections in humans, with encouraging results in terms of bacterial removal and improvement of symptoms.

Overall, this article reviews the recent advances in the discovery and applications of enterococcus-infecting bacteriophages, and points to a few interesting directions for future research in this field.

We thank the reviewer for her/his suggestions and comments to improved our manuscript. The manuscript has been sent to the MDPI English correction service to improve the grammar.

I have the following comments/questions for the authors to address:

  1. The authors need to improve the English writing quite a bit.
  2. that are several where the authors used “ .., being .. “ that are grammatically incorrect. Examples include but are not limited to:

lines 69 70  (…, being some strains proposed as a safe and effective probiotics, ),

It has been changed to: “ the use of some strains as safe and effective probiotics,”

line 149  ( …, being acquired mechanisms the most frequent cause of ampicillin  resistance. ), and

It has been changed to: “ since acquired mechanisms are the most frequent …”

lines 192 193 ( …, being theG2576T (E. coli numbering)  nucleotide mutation the most described ). All of these need to be fixed. For example the last one could be change to …, among which the G2576T (E. coli numbering) nucleotide mutation is the most described.

It has been changed to “among which the G2576T (E. coli numbering) nucleotide mutation is the most described” as suggested.

  1. There are several places where the authors could improve the choice of words. Examples include but are not limited to:

line 93 (“food intoxication” could be change to “food poisoning”)

It has been changed

line 104 (positive could be changed to “beneficial”),

It has been changed

line 127 (“first” could be changed to major ?),

It has been changed

line 168 (“combination” could be changed to ”supplement” ),

“Combination” has been changed to “addition”

line 195 (“lower” could be changed to “minor”),

It has been changed

line 244 (“although” need to be changed to “despite” ).

It has been changed

Line 295 (“responsible” need to be change to “cause”),

It has been changed

line 326 (“suppose” could be changed to “present”),

It has been changed

and line 442 (“a good activity” can be changed to “effective”).

It has been changed

  1. Some sentences seem to be incomplete, which makes them really hard to follow. Examples include but are not limited to:

lines 94 96 In fact, E. faecalis has been identified as the main [pathogen?] responsible for …),

It has been added “species” to clarify the sentence

 line 112 (“Enterococci were subsequently shown [to be the?] cause of several kinds of infections “)

It has been changed

lines 239-240 (“It is remarkably […?]  the number of E. faecalis infecting phages compared with those of E. faecium, …], and

It has been added “highest” and “characterized” to clarify the sentence. It is remarkably […?]  the highest number of E. faecalis infecting phages characterized compared with those of E. faecium, …], and

line 356 357 (Moreover, EFDG1 showed a significantly reduce […?] a 2-week-old biofilm of a vancomycin-resistant E. faecalis strain (V583).”)

It has been changed to “Moreover, EFDG1 showed a significantly reduction of a 2-week-old biofilm of a vancomycin-resistant E. faecalis strain

  1. In section 2.2, the authors might want to point to the readers to previous paper for a more comprehensive review (e.g., Krawczyk et al. Microorganisms 2021).

The suggested review, described the positive and negative effects of Enterococcus strains, so we thought that it would be better to include it in two points of the section 2, Line 73 and L 75.

  1. In section 2.2.2.3, it might be worth mentioning VRE in companion animals and the implications for human health. See for example Wada et al. Antibiotics 2021, 10(2), 138

In this section we are focused on the description of the mechanisms involved in antibiotic resistance in enterococci, we do not think that a reference to companion animals and the risk they suppose for transmission of antibiotic resistance strains will fit in this section.

  1. Lines 266 268 There is a wide range in the genome size distribution, but this heterogeneity is most related to taxonomic differences rather than to genome diversity”): Is it possible to perform ANOVA (or similar) analysis to support this argument?

This sentence is based on the general characteristics that define the taxonomic groups. As it is indicated in the text in the following lines, Herelleviridae phages are based on their capsid morphology but also on their genome size described as long dsDNA from 125-170 Kb while genomes of Rountreeviridae (previously classified as Podoviridae) are small (17-18 kb). We have included references regarding these characteristics (Barylski et al., 2020; Adriaenssens et al., 2020).

  1. Minor suggestions
  2. Line 64: change the citation style of (Sveck and Frank, 2014)

It has been done

  1. Line 96: typo “raisin should be rising”

It has been changed

  1. Line 159: I would suggest replacing “their” with “the bacterial” for clarity. “their” could mean “these agents”.

It has been replaced

  1. Line 170: define VRE (which I assume means vancomycin-resistant Enterococcus) when it first appears in the text.

It has been defined at line 102

  1. Line 238: typo “contest” should be “context”

It has been corrected

Reviewer 2 Report

The present study makes a significant contribution to the field, it is well organized and the table and figure are well presented. The authors present a review, which focuses on showing the advances in the isolation, characterization and application of enterococcal phages as rediscovered weapons against antimicrobial resistant Enterococcus faecium and Enterococcus faecalis that are listed by the WHO for their health problems. Therefore, my decision regarding the manuscript is to approve it as an article as it is.

Author Response

We would like to thank the reviewer for her/his comment

Reviewer 3 Report

  • A brief summary (one short paragraph) outlining the aim of the paper, its main contributions and strengths.

This manuscript provides information about enterococcal phages, their application in food and health, and antibiotic resistance situation in Enterococcus

  • General concept comments

The manuscript provides clear and comprehensive to the field. It starts with the Enterococcus characteristics and antibiotic resistance mechanisms in both E. faecalis and E. faecium. Half of the manuscript donates to the antibiotic resistance mechanisms in both species. Another half provide information on their phages. Even though the authors addressed that the manuscript describes the advances in the isolation, characterization, and application of enterococcal phages as rediscovered weapons against AMR E. faecalis and E. faecium, the advances in the isolation are not obvious. On the other hand, the applications of phage in food and health are well done.

The manuscript provides the up-to-date Enterococcal phages that have been published (supplement). A phylogenetic tree based on phage coat protein was generated to allow more insight into the evolution of enterococcal phages.

Suggestions are:

1. provide the advance in phage isolation

2. discuss more on the phage application

- MOI that has been used (only the example phages)

- burst size

- incubation time (how long does it take to kill the bacterial host)

- concern about using phage in clinical trials (until now, there has been no clinical trial)

- antibiotic resistance and CRISPR in the phage genome

          Specific comment: line 455-456, the phage concentration might be incorrect; please check.

Author Response

This manuscript provides information about enterococcal phages, their application in food and health, and antibiotic resistance situation in Enterococcus

  • General concept comments

The manuscript provides clear and comprehensive to the field. It starts with the Enterococcus characteristics and antibiotic resistance mechanisms in both E. faecalis and E. faecium. Half of the manuscript donates to the antibiotic resistance mechanisms in both species. Another half provide information on their phages. Even though the authors addressed that the manuscript describes the advances in the isolation, characterization, and application of enterococcal phages as rediscovered weapons against AMR E. faecalis and E. faecium, the advances in the isolation are not obvious. On the other hand, the applications of phage in food and health are well done.

The manuscript provides the up-to-date Enterococcal phages that have been published (supplement). A phylogenetic tree based on phage coat protein was generated to allow more insight into the evolution of enterococcal phages.

We would like to thank the reviewer for her/his comments

Suggestions are:

  1. provide the advance in phage isolation

We thank the reviewer for her/his comments. Regarding advance in the isolation of enterococcal phage, there is not more information on new techniques as high-throughput isolation techniques. The most recent advance is the use of alternative sources for phage isolations at has been indicated at L280-284. To avoid any misunderstanding, we have remove isolation from the review focus (L52)

  1. discuss more on the phage application

- concern about using phage in clinical trials (until now, there has been no clinical trial)

This issue has been discussed now in the manuscript (L514-L525)

- MOI that has been used (only the example phages)

- incubation time (how long does it take to kill the bacterial host)

- burst size

Although these type of data is very interesting, we think that it would be more appropriate for a review focused on the description of phages that infect Enterococcus, rather than a review aimed to described their applications, as it is this one. In fact, in many of the manuscripts describing applications these data are not included, especially in the case of human application, in most cases due to the existence of previous works describing the isolation and characterization of these phages. We think that the important conclusion regarding application is if there are some positive results regarding control or elimination of the enterococcal infection.  However, if available in the references, the phage doses applied (either MOI or dose), as well as the phage titter and the time of phage therapy application have been indicated thorough the manuscript (L307, 310, 314,.352-354, 364-370, 378-379, 434-440, 445, 451, 462-466, 468-470, 480-482, 489-491, 497, 508-510).

Antibiotic resistance and CRISPR in the phage genome

As far as we know, there is not description of phage encoded antibiotic resistance genes in the genomes of enterococcus infecting phages. As has been mention in the text, the fact that Enterococcus phage received attention for their application as therapeutic agents, could be a bias for the description of only phage genomes in which such genes were not located, a sentence regarding this aspect has been included in the text (L283-288). Similarly, there are no reference of the presence of CRISPR in the Enterococcus infecting phage genomes. In fact, even in the genome of the host, especially those strains that are described as multi drug resistant lacks CRISPR systems in their genomes (Palmer and Gilmore 2010, mBio. 10.1128/mBio.00227-10).

Specific comment: line 455-456, the phage concentration might be incorrect; please check.

Concentrations of phage used by Pradal et al., and Al-Zubidi et al., in their respective in vivo models have been revised, and they are in concordance with the concentrations indicated in the manuscript.

Round 2

Reviewer 3 Report

This is a good review. I am really enjoy reading the ms. Thanks for all changes.